# Multimodal Depression Detection with Contextual Position Encoding and Latent Space Regularization

## Abstract

Clinical interviews are the gold standard for detecting depression, and previous work using multimodal features from participants' audio, transcribed text, and video have shown promising results. Recent approaches further improve performance by incorporating an additional textual modality—the interviewer's prompts—during training. However, these approaches risk introducing biases, as models may over-rely on specific prompt-response pairs, which may not always be present in real-world settings. This leads to models exploiting these cues as shortcuts for detecting depression rather than learning the language and behaviors that genuinely indicate the subject's mental health, ultimately undermining consistency and objectivity. To address this, we propose a novel approach that combines Contextual Position Encoding (**CoPE**) and Latent Space Regularization (**LSR**), leveraging both subjects' responses (audio) and the interviewer's prompts (text). CoPE captures the evolving context of the interview, ensuring that the model utilizes insights from the entire conversation, preventing over-reliance on isolated or late-stage cues. This helps the model understand interactions holistically and more accurately reflect mental health indicators. LSR introduces constraints to enforce consistency in the model's learned representations, reducing overfitting to superficial cues and guiding the model toward more generalizable patterns. By smoothing the latent space, LSR helps the model focus on meaningful, high-level representations of both audio and text. Our approach yields competitive results on the DAIC-WOZ benchmark and surpasses the state-of-the-art on the EATD benchmark. The code is released[1].

## 1 Introduction

Depression is a widespread condition with severe consequences, such as emotional distress, social withdrawal, and even suicide. According to the World Health Organization (WHO)[2], over 300 million people worldwide suffer from depression, impacting individuals, families, and society profoundly (World Health Organization, 2017). Unfortunately, in many communities, lack of awareness and stigma around mental health problems result in underdiagnosis and undertreatment. This emphasizes the urgent need for practical depression screening tools (Kessler, 2012; Craft & Landers, 1998).

At present, clinical interviews (Figure 1) are still the standard method for evaluating depressive symptoms(He et al., 2022). These interviews mainly consist of four main components: the interviewer's questions, and the participant's responses in text, audio, and video formats, which serve as the basis for supervised learning in automatic depression assessment. Typically, these interviews are semi-structured, multi-round question-and-answer conversations where the interviewer uses a flexible outline to dig into the participant's personal experiences and mental states.

---

[1]https://github.com/icecoldfanta/ICLR2025_depression
[2]https://www.who.int/news-room/fact-sheets/detail/depression

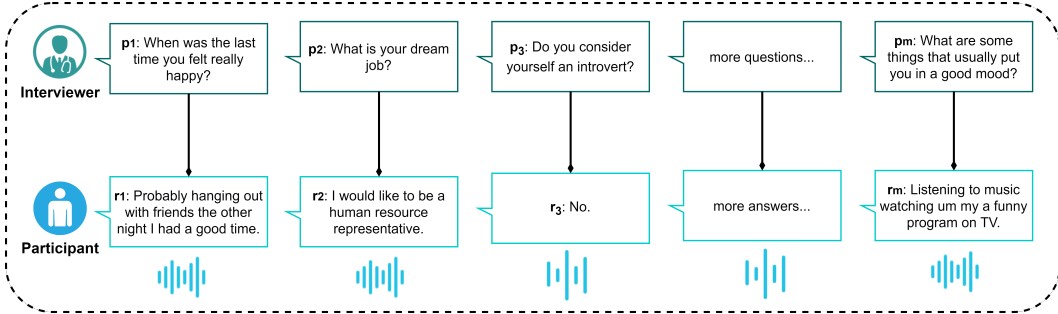

Figure 1: A clinical interview for assessing depression. Each interviewer's prompt is indexed with its position in this conversation flow; each participant's response is aligned with the position of the prompt.

Multimodal approaches that incorporate multiple data sources, such as participants' audio, text, and video collected during interviews, have shown promising results in depression detection. These approaches can be broadly categorized into two types (Al Hanai et al., 2018). The first type models a participant's outcome based on responses to specific questions (e.g., What makes you sad?' or Do you have a history of depression?'), while the second type models outcomes based on responses that are independent of the questions asked (e.g., loudness). In the first approach, (Arroll et al., 2005) and (Burdisso et al., 2024) explored the use of key question sets to optimize screening accuracy while minimizing time. (Yang et al., 2017) combined multiple modalities conditioned on manually selected questions for depression detection. The second approach attempts to exploit global and time-varying statistics, independent of the question that prompted the response, by treating the entire interview (e.g., a waveform file) as a single input data point. For example, (Du et al., 2023) used MFCCs extracted from audio as the primary input, while (Xue et al., 2024) utilized both textual and audio information in their multimodal fusion module.

Recent studies have shown that incorporating the interviewer's prompts as additional textual input, in addition to using participants' audio, video, and transcribed text, can further improve performance (Milintsevich et al., 2023; Chen et al., 2024; Agarwal et al., 2024; Burdisso et al., 2024). However, these methods, along with the first approach discussed above, face two significant challenges. First, focusing on a participant's response to specific mental-state probing questions during an interview may optimize accuracy and reduce screening time, but it may not be practical in real-world scenarios, where participants might try to conceal their true emotional state when asked direct questions (Pretorius et al., 2019; Wilson et al., 2011). Second, conducting clinical interviews consisting of numerous question-answer dialogues, from general questions gradually leading to mental-state-specific questions, is not always feasible. In shorter interviews with random, less predictive questions, previous models tend to be less robust and generalizable.

Given the question-answer format of depression screening tests (Figure 1), we use the interviewer's prompt (text) and the participant's response (audio) as input data for the task of depression detection, framing it as a binary classification problem to predict whether the participant is depressed or non-depressed. We model depression using contextual position encoding (**CoPE**) (Vaswani, 2017; Shaw et al., 2018; Golovneva et al., 2024), which captures the position of interactions within the interview. This allows the model to consider the evolving context of the conversation and ensure that comprehensive information from the entire interview is effectively utilized. This prevents the model from relying on isolated cues that only appear in specific regions of the interview. After combining the data from these two modalities, we apply regularization on the latent space (**LSR**) (Kingma, 2013; LeCun et al., 2015; Higgins et al., 2017) to smooth it by reducing overfitting caused by biases from specific question-answer pairs. This ensures that the model learns consistent patterns.

We evaluated our model on two datasets: DAIC-WOZ (Gratch et al., 2014) and EATD (Shen et al., 2022). Our approach has the potential to enhance automated healthcare decision-making and offer better support for clinicians in practical settings.

## 2 RELATED WORK

### 2.1 SINGLE-MODALITY DEPRESSION DETECTION

(Zhang et al., 2021) proposed DEPA, a self-supervised, pre-trained audio embedding model for depression detection using a traditional encoder-decoder framework. (Nguyen et al., 2022) utilized Reddit data, manually defining PHQ-9-based patterns for classification and introducing a BERT-based model to bypass explicit questionnaires. (Liu et al., 2024) combined contrastive learning and a capsule network to classify depressed and control groups based on PHQ-9 responses. (Du et al., 2023) introduced MSCDR, which used features like MFCC and linear predictive coding with LSTMs to capture dynamic depressive features across speech data. (Burdisso et al., 2023) applied Graph Convolutional Networks with self-connection nodes to capture long-distance semantics in participants' textual responses.

### 2.2 MULTIMODAL DEPRESSION DETECTION

Multimodal approaches combine features from text, audio, and video to enhance depression detection. (Gong & Poellabauer, 2017) used an ensemble of audio, video, and semantic features, with topic modeling to identify relevant interview topics. (Al Hanai et al., 2018) employed LSTMs to model interactions between audio and text features, demonstrating that sequential modeling can detect depression even with minimal structural information. (Wu et al., 2023) leveraged pre-trained foundational models with self-supervised learning, fine-tuning representations from ASR-based emotion and text for classification. The MMPF model(Yang et al., 2024) refined text and audio features using expert knowledge before applying late fusion. (Iyortsuun et al., 2024) introduced a cross-modal attention network (ACMA) with BiLSTMs to capture interactions between audio and text. (Zhang et al., 2024) integrated acoustic landmarks into an LLM framework, converting speech signals into linguistically meaningful tokens for multimodal depression detection.

### 2.3 INCORPORATING INTERVIEWER'S PROMPTS FOR DEPRESSION DETECTION

The Hierarchical Context-Aware Graph (HCAG) attention model proposed by (Niu et al., 2021) applies a Graph Attention Network to integrate contextual information from relational interview questions across audio and text modalities for depression detection. (Dai et al., 2021) extracted topics from interviewer questions and constructed feature vectors combining audio, video, and semantic features for context-aware analysis. To capture interactions between prompts and audio responses, (Shen et al., 2022) utilized GRU and BiLSTM models with an attention layer to summarize multimodal representations. (Milintsevich et al., 2023) framed depression detection as a symptom profile prediction task, training a hierarchical regression model on patient-therapist interview transcripts. (Agarwal et al., 2024) leveraged the interview structure using a multi-view architecture, separating transcripts into patient and therapist views based on sentence type to exploit symmetrical discourse. (Chen et al., 2024) introduced the Structural Element Graph (SEGA), representing clinical interviews—including audio, text, and video—within a directed acyclic graph inspired by expert knowledge, augmented with LLM-based data. (Xue et al., 2024) designed the Channel Attention-based Multimodal Fusion Module (CAMFM), which integrates low-level audio features, wav2vec representations, and BERT-derived sentence embeddings for comprehensive fusion.

(Burdisso et al., 2024) demonstrated that introducing interviewer prompts can bias models to concentrate on certain question-answer pairs. Although some methods take advantage of this by manually selecting predictive pairs, they often lack generalizability, particularly in conversational contexts where specific questions might not be available. To tackle this issue, we propose CoPE, a method that dynamically encodes the position of each interaction within the entire conversation, allowing the model to capture the participant's changing emotional state. Additionally, we regularize the latent space after data fusion to minimize overfitting and ensure that the model learns robust, generalizable patterns.

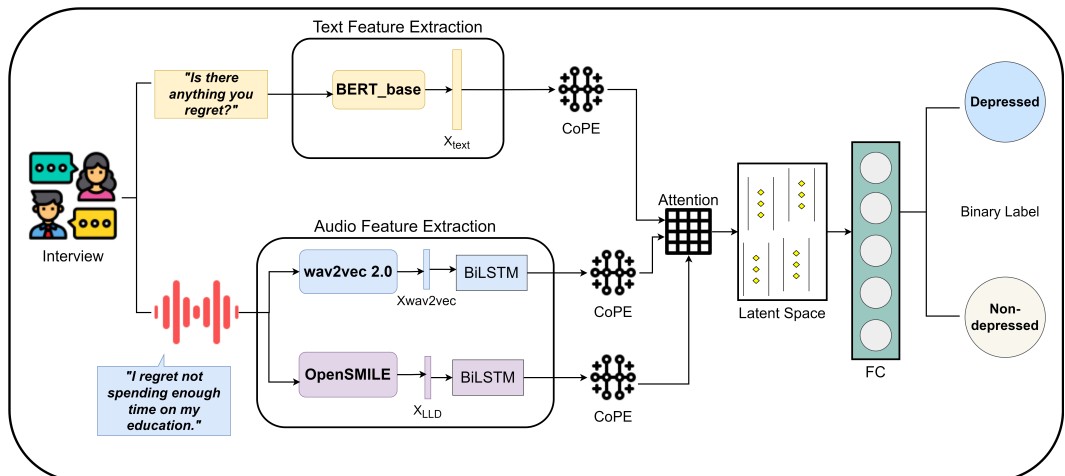

Figure 2: The overall structure of the proposed model.

## 3 METHOD

### 3.1 OVERALL FRAMEWORK AND PROBLEM FORMULATION

In a clinical interview $I$ from the corpus $C$, the interviewer asks the participant a series of $m$ questions (recorded as text) $Q = \{q_1, \ldots, q_m\}$, typically probing into their feelings, experiences, and mental states. The participant's responses are recorded in audios $A = \{a_1, \ldots, a_m\}$ capturing tone and prosody. Each question ($q_i$) and its corresponding audio response ($a_i$) form a single question-response pair.

Given a dataset $D = \{I_1, \ldots, I_n\}$ consisting of $n$ interviews, each interview $I_k$ consists of a set of question-response pairs $\{(q_1, a_1), \ldots, (q_m, a_m)\}$, and the task is to predict whether the participant is depressed (denoted by the binary label $y_k \in \{0, 1\}$). Our proposed overall framework is shown in Figure 2.

Contextual Position Encoding (**CoPE**) combines both absolute position information and contextual information within a sequence to provide a semantically meaningful representation of each interaction (Vaswani, 2017; Golovneva et al., 2024). Unlike absolute position encoding, which treats each position independently, CoPE integrates the surrounding context into the position encoding, allowing the model to simultaneously capture both *token-level* and *sentence-level* positions. This ensures that responses are interpreted not only by their standalone content but also by their position relative to earlier interactions, making it especially suited for tasks like depression detection, where earlier responses may shape later ones.

In our task, CoPE ensures that the model accounts for the evolving nature of the conversation, attending to earlier emotionally charged interactions that may influence subsequent responses. This avoids over-reliance on isolated cues appearing late in the conversation.

We compute the positional encoding $\mathbf{PE}_i$ for each interaction $(q_i, a_i)$ as:

$$\mathbf{PE}_i = \alpha_i \cdot \text{PE}_{\text{absolute}}(i) + \beta_i \cdot \text{PE}_{\text{contextual}}(q_i, a_i, \{q_{<i}, a_{<i}\}) \tag{1}$$

where:

- $\text{PE}_{\text{absolute}}(i)$ encodes the absolute position of interaction $i$ in the sequence.

- $\text{PE}_{\text{contextual}}(q_i, a_i, \{q_{<i}, a_{<i}\})$ incorporates the context of previous question-response pairs.

- $\alpha_i$ and $\beta_i$ are learnable parameters that balance absolute and contextual positional encodings.

This positional encoding is added to the hidden representations of the text and audio for each question-response pair:

$$\mathbf{h}_i^{q,\text{encoded}} = \mathbf{h}_i^q + \mathbf{PE}_i \quad \text{and} \quad \mathbf{h}_i^{a,\text{encoded}} = \mathbf{h}_i^a + \mathbf{PE}_i \tag{2}$$

This ensures that the model interprets each response in the context of prior interactions, helping capture the conversation flow and the evolving emotional state.

Latent Space Regularization (**LSR**) (Kingma, 2013; LeCun et al., 2015; Higgins et al., 2017) is a set of techniques that can be used to reduce domain shift. These techniques have been applied in various semantic segmentation tasks, such as continual learning (Michieli & Zanuttigh, 2021) and few-shot learning (Dong & Xing, 2018; Wang et al., 2019). These strategies generally involve imposing additional constraints on the feature vectors, effectively reducing the space each can occupy. This reduction has promoted more overlap between the source and target distributions, thus decreasing the domain shift (Toldo et al., 2021; Barbato et al., 2021). Unlike traditional weight-based regularization techniques such as $L_1$ or $L_2$, which control model complexity by penalizing large weights, LSR directly constrains the latent feature space. This enables the fused text and audio representations to capture more robust patterns and mitigates overfitting to noise or superficial features, ultimately enhancing model generalization. Without regularization, models can overfit to noisy or superficial patterns, such as specific question-response pairs or keywords that may only appear in certain parts of the interview.

In our approach, LSR is applied to the fused hidden representations of both text and audio after CoPE has been applied. Let $\mathbf{z}_i$ be the fused representation for the $i$-th interaction:

$$\mathbf{z}_i = \text{Fusion}(\mathbf{h}_i^{q,\text{encoded}}, \mathbf{h}_i^{a,\text{encoded}}) \tag{3}$$

To prevent overfitting, we use $L_2$ regularization and dropout on the fused representations. $L_2$ regularization helps discourage large magnitudes in the latent embeddings, while LSR offers additional constraints within the feature space. This alignment of source-target distributions reduces domain shifts, leading to enhanced robustness and better generalization. The regularization loss is defined as:

$$\mathcal{L}_{\text{LSR}} = \lambda \sum_{i=1}^{m} \|\mathbf{z}_i\|_2^2 \tag{4}$$

where:

- $\|\mathbf{z}_i\|_2^2$ is the $L_2$ norm applied to the fused representation to penalize large weights.
- $\lambda$ is the regularization strength parameter.

During training, we also apply Dropout to the fused representations to prevent the model from relying too heavily on any single feature dimension. Dropout randomly sets a fraction of the feature dimensions to zero, implicitly regularizing the model during optimization.

The final objective is to minimize the overall loss, which consists of the standard cross-entropy loss for depression classification and the regularization loss:

$$\mathcal{L} = \mathcal{L}_{\text{CE}} + \mathcal{L}_{\text{LSR}} \tag{5}$$

where $\mathcal{L}_{\text{CE}}$ is the binary cross-entropy loss for predicting whether the participant is depressed or not.

## 3.2 AUDIO FEATURE EXTRACTION

We use two types of audio features: Low-Level Descriptors (LLDs) and wav2vec 2.0 embeddings. LLDs (Eyben et al., 2015) are manually crafted features derived from the raw waveform. They capture both time-domain and frequency-domain properties, such as short-time energy, zero-crossing rate, and spectral centroid, providing a detailed audio signal representation. For LLD extraction,

we use OpenSMILE (Eyben et al., 2010) to generate a 23-dimensional feature set (excluding the bandwidth of the second and third formant peaks), with a frame size of 0.025 seconds and a frame step of 0.01 seconds. Wav2vec 2.0 (Baevski et al., 2020) is a pre-trained model that learns audio representations through multiple convolutional layers. It captures features across varying time and frequency scales, producing high-level embeddings of the raw audio signal.

### 3.3 TEXT FEATURE EXTRACTION

We use BERT (Devlin et al., 2018), a pre-trained transformer model, to generate contextualized text embeddings. Unlike traditional word embeddings, BERT-based sentence embeddings capture intricate word dependencies and contextual nuances, providing a deeper understanding of the relationships within text sequences. This feature makes BERT well-suited for tasks involving rich semantic and syntactic context, such as identifying mental health indicators in interview data (Senn et al., 2022). In our methodology, we employ BERT_base[3] to extract sentence-level embeddings from the interview prompts in the DAIC-WOZ dataset. For the EATD dataset, we utilize a pre-trained Chinese BERT model[4] with the same number of parameters for consistency.

### 3.4 MULTI-MODAL FUSION

To effectively combine audio and text features, we apply a multi-modal fusion strategy that captures complementary information from both modalities. The audio features extracted by *OpenSMILE* and *wav2vec 2.0* and the text features derived from *BERT* are first processed independently using BiL-STM layers to capture temporal dependencies within each modality. The output from the BiLSTM layers is denoted as:

$$h_{\text{audio}}^{\text{BiLSTM}}, h_{\text{text}}^{\text{BiLSTM}}$$

where $h_{\text{audio}}^{\text{BiLSTM}}$ and $h_{\text{text}}^{\text{BiLSTM}}$ represent the hidden states from the audio and text BiLSTMs, respectively.

*CoPE* is then applied to ensure that the model interprets each response in the context of prior interactions, encoding both the absolute and relative positions of each interaction. CoPE dynamically adjusts the positional encoding to reflect the conversation flow:

$$\text{PE}_i = \alpha_i \cdot \text{PE}_{\text{absolute}}(i) + \beta_i \cdot \text{PE}_{\text{contextual}}(i, \{q_{<i}, a_{<i}\}) \tag{6}$$

where $\text{PE}_i$ is the positional encoding for the $i$-th interaction, $\alpha_i$ and $\beta_i$ are learnable weights balancing absolute and contextual encodings, and $q_{<i}, a_{<i}$ are previous questions and answers.

For the fusion process, we employ an *attention mechanism* to selectively focus on the most important components from both the audio and text modalities. This mechanism computes an attention score for each modality, allowing the model to weigh certain features more heavily based on their relevance to the task.

Given the encoded audio and text features $h_{\text{audio}}^{\text{BiLSTM}}$ and $h_{\text{text}}^{\text{BiLSTM}}$, we calculate attention weights using scaled dot-product attention. For Attention$_{\text{audio}}$, the audio features serve as queries ($Q$) and the text features as keys/values ($K, V$). Conversely, for Attention$_{\text{text}}$, the text features serve as queries ($Q$) and the audio features as keys/values ($K, V$). The general formula for attention is:

$$\text{Attention}(Q, K, V) = \text{softmax}\left(\frac{QK^\top}{\sqrt{d_k}}\right) V \tag{7}$$

In our implementation, the attention weights for audio and text are computed as:

$$\text{Attention}_{\text{audio}} = \text{softmax}\left(\frac{h_{\text{audio}}^{\text{BiLSTM}} \cdot (h_{\text{text}}^{\text{BiLSTM}})^\top}{\sqrt{d}}\right) \tag{8}$$

$$\text{Attention}_{\text{text}} = \text{softmax}\left(\frac{h_{\text{text}}^{\text{BiLSTM}} \cdot (h_{\text{audio}}^{\text{BiLSTM}})^\top}{\sqrt{d}}\right) \tag{9}$$

---

[3]https://huggingface.co/google-bert/bert-base-uncased
[4]https://huggingface.co/google-bert/bert-base-chinese

where $d$ is the dimensionality of the encoded features used as keys and queries.

The attention weights are then used to compute weighted representations:

$$h_{\text{audio, weighted}} = \text{Attention}_{\text{audio}} \cdot h_{\text{audio}}^{\text{BiLSTM}} \tag{10}$$

$$h_{\text{text, weighted}} = \text{Attention}_{\text{text}} \cdot h_{\text{text}}^{\text{BiLSTM}} \tag{11}$$

Finally, the fused representation is computed as:

$$h_{\text{fusion}} = \lambda_{\text{audio}} \cdot h_{\text{audio, weighted}} + \lambda_{\text{text}} \cdot h_{\text{text, weighted}} \tag{12}$$

Here, $\lambda_{\text{audio}}$ and $\lambda_{\text{text}}$ are hyperparameters that balance the contributions of each modality.

After fusion, we apply *LSR* to ensure that the learned representations capture generalizable, meaningful patterns while avoiding overfitting to spurious or superficial features. LSR introduces a penalty term that minimizes the variance in the latent space, encouraging smooth and consistent mappings of similar inputs to nearby points in the latent space. Specifically, we define the regularization loss $\mathcal{L}_{\text{LSR}}$ as:

$$\mathcal{L}_{\text{LSR}} = \sum_{i,j} \left\| h_{\text{fusion}}^{(i)} - h_{\text{fusion}}^{(j)} \right\|^2 \cdot \mathbb{I}(y^{(i)} = y^{(j)}) \tag{13}$$

where $h_{\text{fusion}}^{(i)}$ and $h_{\text{fusion}}^{(j)}$ are the fused representations of inputs $i$ and $j$, and $\mathbb{I}(y^{(i)} = y^{(j)})$ is an indicator function that ensures the penalty is applied only when the labels $y^{(i)}$ and $y^{(j)}$ are the same.

This regularization ensures that similar inputs (e.g., interviews from participants with the same depression status) are mapped to nearby points in the latent space, improving the model's ability to generalize across different data points.

The fused and regularized representation $h_{\text{fusion}}$ is then passed through fully connected layers for final binary classification.

## 4 EXPERIMENTS

### 4.1 DATASETS AND IMPLEMENTATION

The DAIC-WOZ dataset (Gratch et al., 2014) is a widely used English dataset consisting of interviews from 189 participants, recorded in transcripts, audio, and video formats. Each participant is labeled with a PHQ-8 score (Kroenke et al., 2009) (ranging from 0 to 24), with participants scoring 10 or higher classified as depressed (D) and those scoring below 10 labeled as control (C). The training set contains data from 107 participants (30 depressed and 77 control), the development set includes 35 participants (12 depressed and 23 control), and the test set has 47 participants. Since the labels for the test set are not publicly available, we follow prior studies and use the development set for evaluation. The dataset is publicly available upon request[5].

The EATD dataset (Shen et al., 2022) is a relatively new Chinese dataset that consists of interviews with 162 student volunteers. These interviews were recorded in both transcript and audio formats. Participants were asked to respond to three randomly selected questions and also complete a Self-rating Depression Scale (SDS) questionnaire. Each participant was then assigned an SDS score (Zung, 1965), with scores of 53 or higher indicating depression. Out of the 162 participants, 132 were found to be non-depressed, while 30 were found to be depressed. The authors have made the dataset publicly available[6].

Data imbalance is common in healthcare-related databases, including the DAIC-WOZ and EATD datasets. To address this during training, we applied resampling to increase the number of depressed samples in the DAIC-WOZ training set. Each clinical interview in DAIC-WOZ contains a minimum of 34 and a maximum of 40 question-answer pairs between the interviewer and the participant. Following previous approaches (Shen et al., 2022; Xue et al., 2024), we selected the first 10 consecutive question-answer pairs $\{(q_1, a_1), \ldots, (q_{10}, a_{10})\}$ from each of the 30 depressed samples to form the first set. However, unlike these approaches, we maintained the original order of responses when

---

[5]https://dcapswoz.ict.usc.edu/
[6]https://github.com/speechandlanguageprocessing/ICASSP2022-Depression

grouping them for resampling to preserve the natural progression of the conversation. The next re-sampled data point from the same depressed participant includes $\{(q_{11}, a_{11}), \ldots, (q_{20}, a_{20})\}$, and then we move to the next participant. We kept this strategy until all 30 original depressed samples were resampled. This approach doubled the number of depressed data points to 60. We continued this process until the number of depressed samples was balanced with the control group at 77 samples. Note that this resampling was only applied to the training set.

| Model | Modality | DAIC-WOZ | | | EATD | | |
|---|---|---|---|---|---|---|---|
| | | F1 Score | Recall | Precision | F1 Score | Recall | Precision |
| CNN | A | **0.86** | **0.85** | **0.87** | 0.74 | 0.72 | **0.75** |
| OpenSMILE+GRU | A | 0.80 | 0.78 | 0.82 | 0.69 | 0.70 | 0.68 |
| wav2vec2.0+GRU | A | 0.78 | 0.74 | 0.81 | 0.66 | 0.65 | 0.67 |
| wav2vec2.0+BiLSTM | A | 0.80 | 0.76 | 0.83 | 0.68 | 0.67 | 0.69 |
| wav2vec2.0+BiLSTM +CoPE+LSR | A | 0.85 | 0.84 | 0.86 | **0.77** | **0.79** | **0.75** |
| BERT+wav2vec2.0+ BiLSTM | A + I | 0.89 | 0.88 | 0.91 | 0.81 | 0.84 | 0.79 |
| BERT+OpenSMILE+ BiLSTM | A + I | 0.84 | 0.83 | 0.85 | 0.74 | 0.72 | 0.75 |
| GRU+wav2vec2.0 | A + I | 0.81 | 0.78 | 0.83 | 0.70 | 0.68 | 0.72 |
| Transformer+wav2vec2.0 | A + I | 0.85 | 0.84 | 0.86 | 0.76 | 0.77 | 0.74 |
| Full model | A + I | **0.92** | **0.91** | **0.92** | **0.84** | **0.87** | **0.85** |

Table 1: The results from baseline approaches on the DAIC-WOZ and EATD datasets are presented here. In this context, *A* represents the audio information collected from participants, while *I* refers to the interviewer's prompts, which are provided in text form. Only the models that include the 'CoPE' component have this method applied, except for the 'Full model,' which is our complete model and is highlighted in blue. The best results are indicated in **bold**, while the second-best results are underlined.

For the EATD dataset, we followed the resampling strategy proposed by the original authors (Shen et al., 2022), which involves reordering and combining responses to the three questions answered by depressive participants. Since the three questions posed by the interviewer are random, there are six possible rearrangements of the question-response pairs, thereby increasing the size of the depressed class in the training set by a factor of six. The EATD dataset is divided into training and development sets using 3-fold cross-validation, with a 2:1 ratio between the training and development data.

After resampling, we have 154 and 208 audio samples for training from the DAIC-WOZ and EATD datasets, respectively. Since the current version of DAIC-WOZ lacks interview videos, and the EATD does not contain video at all, visual cues are not utilized in this study. Please find detailed statistics of the two datasets in Appendix A.2, Table 4, and our discussion on the limitations of these datasets in Appendix A.10. To ensure consistency, each question-answer pair within an audio sample—10 pairs per sample for DAIC-WOZ and 3 pairs per sample for EATD—is adjusted to fixed durations of 3 seconds and 10 seconds, respectively. This standardization ensures that all audio samples from both datasets have a consistent length of 30 seconds.

In line with most previous works, we use the F1 score, accuracy, and precision as evaluation metrics for our binary classification task. The rationale and further details on these metrics can be found in Appendix A.7. Our analysis and training were executed on a PC running Windows 10 (version 10.0.19041) with an AMD64 architecture and an Intel 64 processor. The system utilized an NVIDIA GeForce RTX 2070 GPU with driver version 536.23 and CUDA version 12.2. The system had 32 GB of RAM and the GPU had 8 GB of VRAM.

## 4.2 COMPARISON WITH BASELINES

We have categorized the baselines (Table 1) into two groups: **(1) Unimodal approaches:** These methods use only the participant's audio as input. We do not include methods that use transcribed text from participants since our model does not incorporate this information. We adopt four methods

| Study | Modality | DAIC-WOZ | | | EATD | | |
|-------|----------|----------|------|-----------|------|------|-----------|
| | | F1 Score | Recall | Precision | F1 Score | Recall | Precision |
| DEPA (Zhang et al., 2021) | A | 0.94 | **0.94** | 0.94 | – | – | – |
| GCN (Burdisso et al., 2023) | T | 0.84 | – | – | – | – | – |
| MSCDR (Du et al., 2023) | A | 0.74 | 0.67 | 0.67 | – | – | – |
| GONG (Gong & Poellabauer, 2017) | A + T + V | 0.80 | – | – | – | – | – |
| HANAI (Al Hanai et al., 2018) | A + T | 0.77 | 0.83 | 0.71 | – | – | – |
| WU (Wu et al., 2023) | A + T | 0.89 | – | – | – | – | – |
| MMPF (Yang et al., 2024) | A + T | 0.88 | 0.93 | 0.83 | – | – | – |
| ACMA (Iyortsuun et al., 2024) | A + T | 0.82 | 0.86 | 0.79 | 0.70 | 0.70 | 0.65 |
| LLM (Zhang et al., 2024) | A + T | 0.83 | – | – | – | – | – |
| CAMFM (Xue et al., 2024) | A + T | 0.92 | 0.92 | 0.92 | 0.77 | 0.84 | 0.73 |
| DAI (Dai et al., 2021) | A + T + V + I | **0.96** | 0.92 | **1.0** | – | – | – |
| HCAG (Niu et al., 2021) | A + T + I | 0.92 | 0.92 | 0.92 | – | – | – |
| SHEN (Shen et al., 2022) | A + I | 0.85 | 0.92 | 0.79 | 0.71 | 0.84 | 0.62 |
| MILI (Milintsevich et al., 2023) | T + I | 0.80 | – | – | – | – | – |
| SEGA (Chen et al., 2024) | A + T + V + I | 0.88 | – | – | 0.81 | – | – |
| BURD (Burdisso et al., 2024) | T + I | 0.90 | – | – | – | – | – |
| AGAR (Agarwal et al., 2024) | T + I | 0.77 | – | – | – | – | – |
| **Our approach** | A + I | 0.92 | 0.91 | 0.92 | **0.84** | **0.87** | **0.85** |

Table 2: Experimental results obtained by training and testing on the DAIC-WOZ and EATD datasets separately. *A*, *T*, and *V* represent audio, transcribed text, and visual information collected from participants, respectively. *I* denotes the interviewer's prompt, provided in text form. – indicates unavailable data. The best results are shown in **bold**, while the second-best results are underlined.

for audio data: CNN, OpenSMILE + GRU, wav2vec + GRU, and wav2vec + BiLSTM. **(2) Multimodal methods** use the participant's audio and the interviewer's question as input. In this group, we compare four baselines: wav2vec + Transformer encoder for text, wav2vec + GRU, BERT and OpenSMILE with BiLSTM for each modality, and BERT with wav2vec 2.0, with BiLSTM for each branch. Please refer to Appendix A.5 for more implementation details.

### 4.3 COMPARISON WITH STATE-OF-THE-ART

We compared our method with three groups of state-of-the-art (SOTA) methods in Table 2: (1) **Unimodal methods**, which use either participants' audio or transcribed text only; (2) **Multimodal approaches**, which incorporate two or more modalities from participants; and (3) **Multimodal approaches that also consider the interviewer's question as an additional modality**.

Our approach achieved competitive performance across all metrics on the DAIC-WOZ dataset and state-of-the-art performance on the EATD dataset. Compared to unimodal approaches, most multimodal methods demonstrated higher overall F1 score, recall, and precision, highlighting the value of leveraging multiple modalities for more comprehensive depression detection. The superior perfor-

| Model Variant | DAIC-WOZ | | | EATD | | |
|---|---|---|---|---|---|---|
| | F1 Score | Recall | Precision | F1 Score | Recall | Precision |
| Full Model | 0.92 | 0.91 | 0.92 | 0.84 | 0.87 | 0.85 |
| Ablate CoPE | 0.88 | 0.87 | 0.88 | 0.83 | 0.85 | 0.82 |
| Ablate LSR | 0.89 | 0.88 | 0.90 | 0.82 | 0.84 | 0.81 |
| Ablate CoPE and LSR | 0.85 | 0.84 | 0.85 | 0.78 | 0.79 | 0.77 |
| Audio Only (wav2vec + BiLSTM) | 0.80 | 0.79 | 0.81 | 0.73 | 0.72 | 0.74 |
| Simple Fusion | 0.89 | 0.88 | 0.90 | 0.82 | 0.84 | 0.81 |

Table 3: Ablation study results for DAIC-WOZ and EATD datasets, comparing F1 Score, Recall, and Precision.

mance of our approach on the EATD dataset underscores the importance of considering the position of interactions. When integrating the interviewer's prompt, the consistently high performance on the DAIC-WOZ dataset suggests a potential bias, as (Burdisso et al., 2024) verified, which may lead to overfitting.

### 4.4 ABLATION STUDY

In Table 3, we conducted experiments to analyze the impact of certain design components discussed in Section 3.1. The first three rows of Table 3 present the results of removing CoPE, LSR, and both, respectively. Removing CoPE led to decreased performance on both datasets. However, the drop was more significant for the DAIC-WOZ dataset, which emphasizes the importance of conversational structure in this dataset, compared to the EATD dataset, where random questions were asked.

When LSR was removed, the performance on DAIC-WOZ was better than when CoPE was ablated, suggesting that the model may rely more heavily on specific features, leading to poorer generalization. Furthermore, the performance decreased further when both CoPE and regularization were ablated.

The unimodal approach resulted in an expected performance drop, as using only audio provides less contextual understanding. Finally, removing the attention mechanism used to fuse audio and text and instead simply concatenating the text and audio embeddings resulted in slightly worse performance than the proposed approach. This indicates that attention-based fusion provides better integration of multimodal information for this task. To explore the datasets used in this study further, we conducted additional experiments outlined in Appendix A.8.

## 5 CONCLUSIONS

This paper presents a method for identifying depression through interviews by combining participants' audio and interviewers' text prompts. To overcome the limitations of prior methods reliant on specific question-answer pairs, we introduce two components: Contextual Position Encoding (CoPE), which captures the evolving nature of interactions, and Latent Space Regularization (LSR), which promotes robust, generalizable representations while reducing overfitting.

Our method achieved competitive performance on all metrics on the DAIC-WOZ dataset and state-of-the-art performance on the EATD dataset. By focusing on contextually relevant interactions and capturing the flow of the conversation, we significantly improved the detection of subtle emotional patterns associated with depression. Our ablation study demonstrated that our approach is more resilient when faced with datasets containing random, non-conversational style data. Future work could explore additional modalities, such as visual cues or physiological signals, to enhance the model's predictive power. Additionally, extending the framework to other mental health conditions could provide valuable insights into how different features contribute to various mental states. Our approach offers a promising direction for building more consistent and generalizable systems for clinical applications. The limitations of our work are discussed in Appendix A.10.

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

| Dataset | Before resampling | | | | After resampling | | | |
|---------|------|-----|-------|-------------|------|-----|-------|-------------|
|         | D    | C   | Train | Development | D    | C   | Train | Development |
| DAIC-WOZ | 30  | 77  | 107   | 35          | 77   | 77  | 154   | 35          |
| EATD     | 30  | 132 | 108   | 54          | 180  | 132 | 208   | 104         |

Table 4: Detailed statistics for the DAIC-WOZ and EATD datasets are provided both before and after resampling. '**D**' represents the depressed group, while '**C**' denotes the control (non-depressed) group.

## A  APPENDIX

### A.1  OVERVIEW

In this supplement, we will first provide data preprocessing steps performed by the original authors in Appendix A.2. Next, we describe the low-level descriptors extracted using OpenSMILE in Appendix A.3. Following that, we present the details of our hyperparameter selection for each module in Appendix A.4, along with the implementation details of the baselines in Appendix A.5. Finally, we discuss the limitations of our wrok in Appendix A.10.

### A.2  DATA PREPROCESSING

Several preprocessing operations were carried out in the EATD dataset (Shen et al., 2022). Initially, mute audios, recordings shorter than 1 second, and silent segments at the beginning and end of each recording were removed. Using RNNoise (Valin, 2018) eliminated background noise with default settings. Although not utilized in our study, (Povey et al., 2014) was employed to transcribe the audio.

For the DAIC-WOZ dataset (Gratch et al., 2014), the audio was recorded using a high-quality close-talking microphone and then transcribed using the ELAN tool from the Max Planck Institute for Psycholinguistics (Brugman et al., 2004). Each transcription was manually reviewed for accuracy, and utterances were segmented at boundaries with at least 300 milliseconds of silence.

### A.3  LLDS

- **Frequency-related (Pitch) features**:
  - Pitch (fundamental frequency, F0)
- **Loudness and energy features**:
  - Loudness
  - Shimmer (local)
- **Spectral features**:
  - Harmonics-to-noise ratio (HNR)
  - Formant 1 (frequency)
  - Formant 1 (bandwidth)
  - Formant 2 (frequency)
  - Formant 2 (bandwidth)
  - Formant 3 (frequency)
  - Formant 3 (bandwidth)
- **Spectral shape features**:
  - Alpha ratio (50–1,000 Hz / 1,000–5,000 Hz energy ratio)
  - Hammarberg index (0–2,000 Hz / above 2,000 Hz energy ratio)
  - Spectral slope (0–500 Hz)
  - Spectral slope (500–1,500 Hz)
  - Spectral flux

- – Spectral centroid
- – Formant energy ratio (F1 to F3)

- **Voice quality (glottal) features**:
    - – Relative energy of harmonics below 500 Hz
    - – Relative energy of harmonics below 1,000 Hz
    - – Jitter (local)
    - – Logarithmic harmonic-to-noise ratio (HNR)

- **Temporal (duration-related) features**:
    - – Voiced segment duration
    - – Unvoiced segment duration
    - – Speech rate

## A.4 DETAILS OF HYPERPARAMETER SELECTION

**BERT:** We use a pre-trained BERT model (base version) for text feature extraction, with an embedding dimension of 768. The maximum sequence length is set to 128 tokens to capture the participant's text responses. The learning rate is set to $3 \times 10^{-5}$, with a batch size of 16. AdamW optimizer is used for training with a dropout rate of 0.1. We freeze the first 12 transformer layers of BERT and fine-tune the last 4 layers.

**wav2vec 2.0:** For audio feature extraction, we use the pre-trained wav2vec 2.0 (base) model, producing embeddings with a dimension of 768. The learning rate is set to $3 \times 10^{-5}$, with a batch size of 16. The optimizer used is AdamW, and a dropout of 0.1 is applied. We freeze the first 12 layers of wav2vec 2.0 and fine-tune the last 2 layers to capture specific audio features relevant to our task.

**OpenSMILE:** We use the OpenSMILE toolkit to extract Low-Level Descriptors (LLDs) from the audio responses. The frame size is set to 0.025 seconds with a frame step of 0.01 seconds. The extracted LLD feature set is based on the GeMAPS (Eyben et al., 2015) configuration, consisting of 23 dimensions. The default energy and zero-crossing rate thresholds are used for extracting these features.

**BiLSTM:** Both audio and text features are passed through separate BiLSTM networks. For the wav2vec 2.0 features, the input dimension is 768, and for the OpenSMILE features, it is 23. The BERT embeddings also have an input dimension of 768. Each BiLSTM has a hidden dimension of 256, with 2 layers and a bidirectional configuration. Dropout of 0.3 is applied, and the activation function used is ReLU. The learning rate for the BiLSTMs is set to $1 \times 10^{-4}$ with the Adam optimizer.

**Contextual Position Encoding (CoPE):** CoPE is applied to both audio and text features, with an input dimension of 768. The positional encoding dimension is set to 512. Learnable weights $\alpha$ and $\beta$ are initialized randomly. The positional encoding method uses cumulative gates to capture both absolute and contextual information. The learning rate is set to $5 \times 10^{-5}$ with a dropout of 0.2, and the optimizer used is AdamW.

**Attention Mechanism (Fusion):** To fuse the multimodal embeddings, we use a scaled dot-product attention mechanism. The input dimension for the fused audio and text features is 768, with 8 attention heads. Dropout is set to 0.1, and the learning rate is $2 \times 10^{-5}$. The output dimension after fusion is 768. AdamW is used as the optimizer.

**Latent Space Regularization (LSR):** To prevent overfitting, we apply $L_2$ norm regularization with a strength of 0.01 to the fused latent space, which has a dimension of 768. Dropout of 0.3 is applied to the latent space. The batch size is set to 32, with a learning rate of $1 \times 10^{-4}$, and the optimizer used is Adam.

**Fully Connected (FC) Layer:** The final fused representations are passed through a fully connected layer for binary classification. The input dimension is 768, with a hidden layer of 128 dimensions. The activation function is ReLU, and dropout is set to 0.2. The output dimension is 1 (binary: depressed vs. non-depressed). The learning rate for the FC layer is $1 \times 10^{-4}$, and AdamW is used for optimization.

## A.5 DETAILS OF BASELINE IMPLEMENTATION

**wav2vec 2.0 + BiLSTM:** This model utilizes wav2vec 2.0 to generate 768-dimensional speech embeddings from raw audio. These embeddings are then processed by a BiLSTM with 256 hidden units to capture temporal dependencies, followed by a fully connected layer for binary classification.

**OpenSMILE + GRU:** OpenSMILE is used to extract low-level descriptors (LLDs). The extracted features are then fed into a GRU with 128 hidden units to capture temporal patterns, followed by a fully connected layer for classification.

**wav2vec 2.0 + GRU:** The 768-dimensional embeddings from wav2vec 2.0 go through a 2-layer GRU with 256 hidden units per layer to capture temporal structure. The output is then passed through a fully connected layer for classification.

**CNN:** A CNN processes the raw audio signal directly using 4 convolutional layers with ReLU activations and max-pooling. After a global average pooling layer, the features are passed through a fully connected layer for classification.

**BERT + wav2vec 2.0 + BiLSTM:** In this model, BERT processes the text input (interviewer's prompts), while wav2vec 2.0 handles audio. Both sets of embeddings are processed by BiLSTMs with 256 hidden units per modality, fused with an attention mechanism, and passed through a fully connected layer for classification.

**BERT + OpenSMILE + BiLSTM:** Here, BERT processes the text input, and OpenSMILE extracts LLDs from the audio. Both sets of outputs are processed by 2-layer BiLSTMs with 256 hidden units per layer, and their outputs are concatenated and passed through a fully connected layer for final classification.

**GRU (Text) + wav2vec 2.0:** A GRU with 128 hidden units processes tokenized text, while wav2vec 2.0 extracts 768-dimensional audio embeddings. The outputs are concatenated and passed through a fully connected layer for binary classification.

**Transformer Encoder (Text) + wav2vec 2.0:** In this model, a 6-layer transformer encoder with 8 attention heads processes tokenized text sequences. wav2vec 2.0 extracts audio embeddings, and both outputs are fused using an attention mechanism. The fused embeddings are classified using a fully connected layer.

## A.6 EXTENDED ABLATION ANALYSIS

In this section, we present additional ablation studies comparing the performance of LSR with traditional weight-based regularization techniques, namely $L_1$ and $L_2$ regularization. The aim of this comparison is to highlight the contributions of LSR to our proposed framework and to assess its effectiveness relative to commonly used weight-based regularization methods.

| Method | F1 Score | Precision | Recall |
|---|---|---|---|
| $L_1$ Regularization Only | 0.82 | 0.80 | 0.84 |
| $L_2$ Regularization Only | 0.85 | 0.83 | 0.86 |
| LSR (our approach) | 0.92 | 0.92 | 0.91 |

Table 5: Comparison of LSR with traditional weight-based regularization techniques.

We conducted three additional experiments to provide a comparative analysis:

1. $L_1$ **Regularization Only**: Applied directly to the model weights using the loss function:

$$\mathcal{L}_{\text{L1}} = \lambda \sum_{j=1}^{n} |\mathbf{w}_j|$$

where $\mathbf{w}_j$ represents the model weights and $\lambda$ is the regularization strength parameter.

2. $L_2$ **Regularization Only**:

$$\mathcal{L}_{\text{L2}} = \lambda \sum_{j=1}^{n} \mathbf{w}_j^2$$

This regularization term penalizes large weights to prevent overfitting.

3. **LSR Combined with $L_2$ Regularization (Baseline)**: This configuration aligns with our main model, where LSR is applied to the fused latent representations alongside $L_2$ regularization on the model weights, with the overall regularization loss defined as:

$$\mathcal{L}_{\text{LSR}} = \lambda \sum_{i=1}^{m} \|\mathbf{z}_i\|_2^2$$

Here, $\mathbf{z}_i$ represents the fused latent representation for the $i$-th interaction, and the $L_2$ norm encourages smaller embeddings.

The results presented in Table 5 demonstrate that incorporating LSR with $L_2$ regularization significantly outperforms purely weight-based approaches. Specifically:

- $L_1$ **Regularization Only**: While $L_1$ regularization promotes sparsity by reducing the impact of less important weights, it does not capture the complex interactions inherent in the fused text and audio data, resulting in comparatively lower performance.
- $L_2$ **Regularization Only**: $L_2$ regularization effectively controls the magnitude of model weights, providing moderate improvements over $L_1$. However, it lacks the constraints imposed by LSR on the latent feature space, which are crucial for effective domain adaptation and feature alignment.
- **LSR Combined with $L_2$ Regularization**: By regularizing the fused latent representations, LSR promotes robust feature learning and reduces domain shift, yielding significant improvements in model generalization and robustness.

These findings emphasize the complementary role of LSR in enhancing the quality and interpretability of fused representations, justifying its inclusion within our proposed framework.

## A.7 RATIONALE FOR METRIC SELECTION

In this study, we used F1 score, precision, and recall as evaluation metrics due to their status as widely recognized standards for binary classification tasks. Additionally, these metrics facilitate a fair and meaningful comparison with previous studies in the field, as shown in Table 2.

In our analysis: - True positives (**TP**) refer to participants correctly predicted as depressed. - False positives (**FP**) refer to participants incorrectly predicted as depressed. - False negatives (**FN**) refer to participants who are actually depressed but incorrectly classified as non-depressed by the model.

The metrics are defined as follows:

- **Precision** measures the proportion of true positive predictions among all positive predictions made by the model. It is defined as:

$$\text{Precision} = \frac{\text{TP}}{\text{TP} + \text{FP}} \tag{14}$$

- **Recall** (also known as sensitivity) measures the proportion of true positive predictions among all actual positive instances. It is defined as:

$$\text{Recall} = \frac{\text{TP}}{\text{TP} + \text{FN}} \tag{15}$$

- **F1 Score** is the harmonic mean of precision and recall, providing a balanced measure of both. It is defined as:

$$\text{F1 Score} = 2 \times \frac{\text{Precision} \times \text{Recall}}{\text{Precision} + \text{Recall}} \tag{16}$$

## A.8 IMPACT OF INTERVIEW LENGTH

To assess the effectiveness of CoPE and LSR with varying session lengths, we performed cross-domain experiments on the DAIC-WOZ dataset (which includes longer sessions, as shown in Figure 3) and the EATD dataset (which provides for shorter sessions, as shown in Figure 4). For these

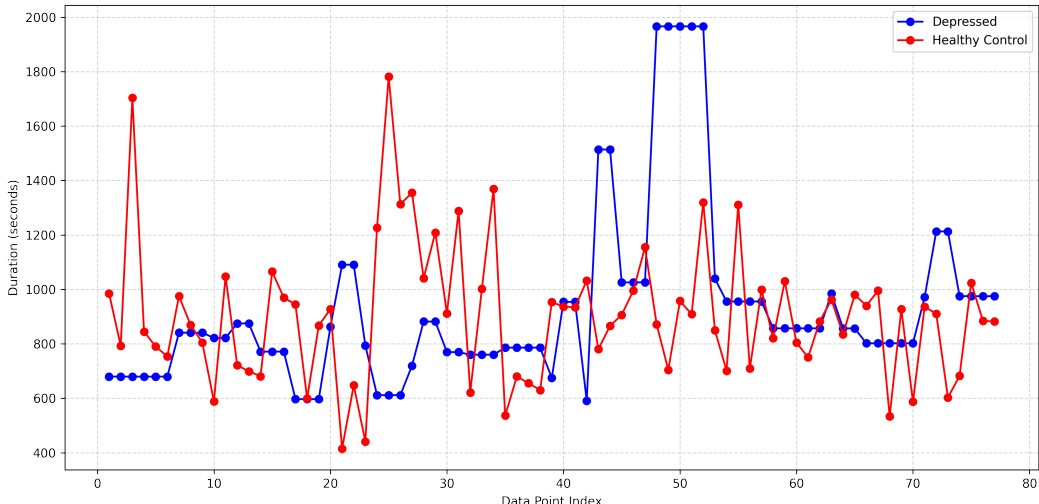

Figure 3: From the DAIC-WOZ dataset, the average duration of interviews labeled as depressed is 928.28 seconds, whereas the average for healthy controls is 904.07 seconds. This results in an average interview duration of 916.17 seconds across the entire dataset.

experiments, we utilized raw data, applying oversampling (as discussed in Section 4.1) for class balancing but without fixing the input length. The results, presented in Table 6, highlight CoPE's effectiveness in handling sessions of different lengths.

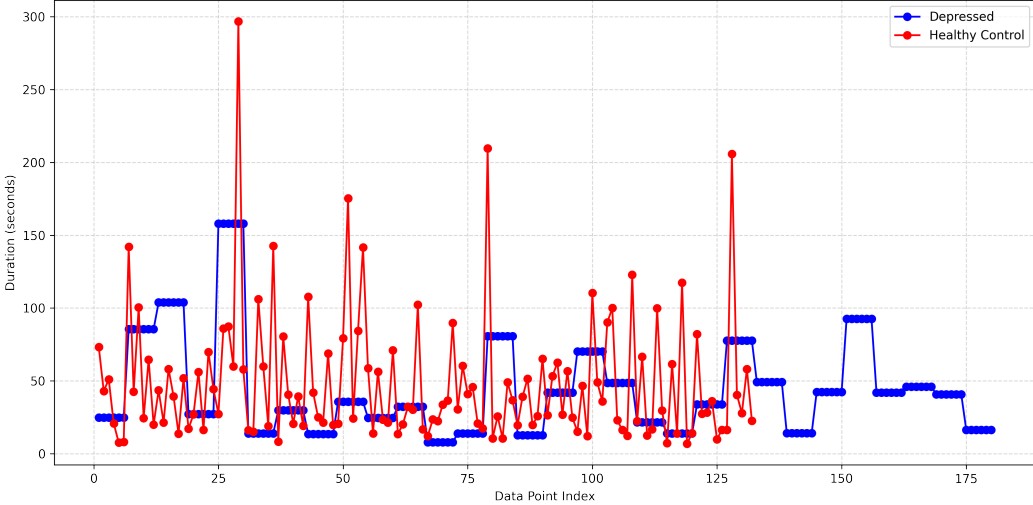

Figure 4: The duration of each interview session in the EATD dataset varied. The mean duration for interviews labeled as depressed was 43.86 seconds, while for healthy controls, it was 48.72 seconds. This results in an average duration of 45.92 seconds across the whole dataset.

The significant decrease in performance observed when CoPE is removed, especially in experiments using the DAIC-WOZ dataset for both training and testing, highlights CoPE's crucial role in capturing long-term contextual dependencies in extended dialogues. Longer sessions naturally offer richer and more complex contextual information, allowing CoPE to effectively encode sequential and cross-turn relationships, which leads to considerable performance improvements.

In the EATD dataset, which consists of shorter sessions, CoPE still provides a positive contribution, but to a lesser extent. The reduced context in these brief interactions limits CoPE's ability to utilize

| Train | Test | F1 | Recall | Precision | F1-Ablate CoPE | F1-Ablate LSR |
|-------|------|----|--------|-----------|----------------|---------------|
| DAIC-WOZ | DAIC-WOZ | 0.89 | 0.87 | 0.90 | 0.82 | 0.83 |
| DAIC-WOZ | EATD | 0.75 | 0.76 | 0.74 | 0.69 | 0.70 |
| EATD | EATD | 0.82 | 0.83 | 0.81 | 0.75 | 0.76 |
| EATD | DAIC-WOZ | 0.70 | 0.71 | 0.69 | 0.64 | 0.66 |

Table 6: Cross-corpus validation results between DAIC-WOZ and EATD datasets.

long-term dependencies effectively. Thus, while CoPE is beneficial in both datasets, its effectiveness is more pronounced in cases with longer sessions, with more abundant contextual information available.

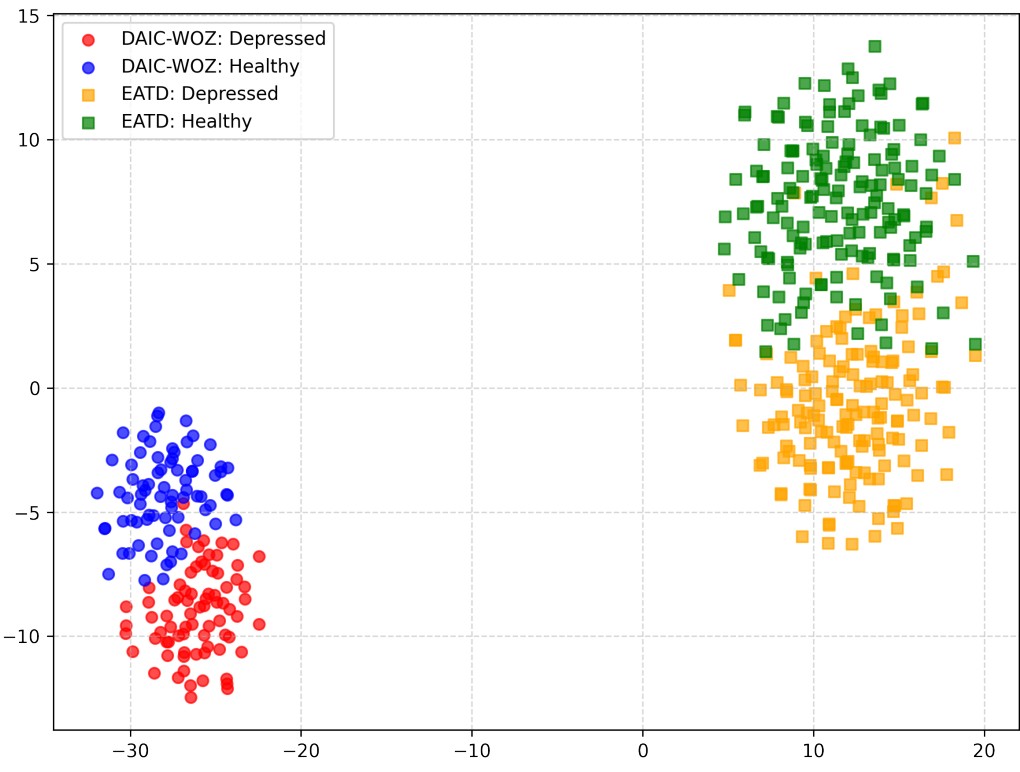

Figure 5: Visualization of fused multimodal features for the DAIC-WOZ and EATD datasets.

To evaluate the quality of the learned feature representations, we reduced the high-dimensional vectors to a two-dimensional plane using t-SNE (Van der Maaten & Hinton, 2008). As shown in Figure 5, distinct clusters were formed, clearly separating depressed and healthy participants across both datasets, which demonstrates excellent separability. This indicates that combining CoPE and attention mechanisms effectively integrates multimodal information, resulting in meaningful and distinguishable representations.

## A.9 STATISTICAL SIGNIFICANCE TESTING

To evaluate the robustness of our results and address concerns regarding dataset size, we conducted statistical significance testing on the performance metrics of our model in comparison to baseline

| Dataset | Paired t-test (p-value) | Wilcoxon signed-rank test (p-value) |
|---------|------------------------|--------------------------------------|
| DAIC-WOZ | 0.01 | 0.02 |
| EATD | 0.03 | 0.04 |

Table 7: Results of statistical significance testing for F1 scores of the proposed model versus baselines.

approaches. Specifically, we performed paired t-tests and Wilcoxon signed-rank tests on the F1 scores obtained from multiple experimental runs.

For the DAIC-WOZ dataset, we followed standard practices by evaluating the provided development set. We repeated the experiments with different random seeds five times and used these results for statistical significance testing. For the EATD dataset, we applied five-fold cross-validation, which was consistent with prior studies. Performance metrics were computed for each fold, and the aggregated results from the five folds were subsequently used for statistical significance testing. These protocols align with prior work and provide a fair basis for evaluating statistical significance.

We used paired t-tests to assess the significance of performance differences between our proposed model and the most robust baseline model (BERT + wav2vec2.0 + BiLSTM). The results are summarized in Table 7.

The p-values indicate that the performance improvements of our proposed model over the baseline are statistically significant on F1 for both datasets ($p < 0.05$) . This validates the robustness of our approach, even when working with relatively small datasets. The observed differences are not attributable to random chance, underscoring the performance of our method.

### A.10 LIMITATIONS

Our work has several limitations. First, although we integrate audio and text modalities through an attention-based fusion mechanism, we do not include other potentially useful modalities such as visual cues (e.g., facial expressions or gestures), which have been shown to provide relevant signals in assessing mental health conditions. There are two main reasons for not including visual cues or physiological data in this study. The first reason is the availability of such data in real-world settings. Labeled speech data is limited, and clinically labeled speech data is even scarcer. Therefore, to make a framework that can be integrated into real-world clinical scenarios, we aimed to develop a framework that operates efficiently with easily accessible data, specifically the patient audio recordings and the questions posed by the clinician or interviewer. The second reason is that we used the DAIC-WOZ and EATD datasets in this study because both of them are interview-based. However, the EATD dataset does not include video recordings, preventing us from utilizing visual cues in our analysis. We acknowledge that incorporating visual or physiological data could further improve the accuracy and robustness of the model in future work, provided that more datasets with comprehensive modalities of data become available.

Several critical factors must be considered to implement our model in real-world clinical settings. First, even when data is collected through interviews, variations such as language differences, the types of speech tasks (e.g., reading scripted sentences, engaging in spontaneous speech, or recall tasks), and the recording environments (e.g., laboratory settings, natural settings, one-on-one interactions, or group interactions) can significantly affect the downstream task of depression detection. Second, when data is not collected through interviews, capturing and modeling the complex patterns between an interviewer's prompts and a participant's responses becomes more challenging. Therefore, in our future work, we aim to develop a more generalizable and reliable approach to ensure our model is better suited for real-world applications.

Additionally, while effective at handling imbalanced datasets through LSR, our approach still heavily relies on the availability of annotated data, which can be costly and time-consuming to collect in clinical settings. The use of semi-supervised or unsupervised techniques could alleviate this reliance on labeled data and further enhance the model's generalizability.

