# OpenReview forum: "Multimodal Depression Detection with Contextual Position Encoding and Latent Space Regularization"
_ICLR.cc/2025/Conference — Submitted to ICLR 2025_

### Official Review · Reviewer_eyjC · 2024-10-28

**Soundness:** 2
**Presentation:** 2
**Contribution:** 2
**Rating:** 3
**Confidence:** 4

**Summary:**

This paper provides a method for detecting depression in clinical interviews by combining Contextual Position Encoding and Latent Space Regularization. This involves tracking the context as it evolves over the conversation, and subsequently smoothing latent representations. This is meant to better capture the ‘flow’ of the conversation and avoid overfitting. Although this provides good results on DAIC-WOZ and EATD, additional modalities (like video) could have been pursued.

**Strengths:**

- the more contextual approach of CoPE is a good addition to the (varied and numerous) approaches to this problem. This might be a better way to obtain pragmatic cues or more speaker-specific baselines than focusing on cross-sectional aspects across speakers. Using LSR to smooth the latent space is also a correct step to take, although there are alternatives. This is, on the whole, positive but fairly evolutionary.
- SotA performance is, of course, reported on this important task. Ablations are run on the appropriate parts of the model in Table 3, and various alternatives are mentioned in Table 2 for comparison.

**Weaknesses:**

- While it is positive that DAIC-WOZ and EATD were used, this is still a somewhat reduced and well-trodden set. They are also fairly curated and may not reflect real-world variability or noise, which limits claims of generalization or robustness.
- Despite the (almost necessary) ablations, the evaluations are fairly superficial, focusing on precision, recall, and F1 but not going deeper into the results to vary the parameters further, or to explain them, or to identify patterns in performance on aspects of the dataset. The result is a fairly ‘paint-by-numbers’ set of experiments with unclear gain in knowledge.

**Questions:**

- Could you check your references for, e.g., completeness (e.g., how the WHO 2017 paper was published) or formatting?
- What real-world aspects may be relevant to consider for additional analysis? E.g., how might environmental considerations or more unstructured data collection relate to your results? What are the limitations of this work that you are able to identify?

---

> ### Author Response · Authors · 2024-11-21
> **Author Response to Reviewer eyjC**
>
> Dear Reviewer eyjC,
>
> Thank you for raising concerns about the dataset and sharing your insightful thoughts on generalizability. Please see our response below to each concern raised.
>
> **Question 1:**
> > While it is positive that DAIC-WOZ and EATD were used, this is still a somewhat reduced and well-trodden set. They are also fairly curated and may not reflect real-world variability or noise, which limits claims of generalization or robustness.
>
> Thank you for this thoughtful observation. We agree that the conditions under which speech data is collected—whether in the lab, in the wild, through reading tasks, structured interviews, or spontaneous speech—play a significant role in downstream performance, as discussed in Section A.10 of the Appendix. This inherent variability is a common challenge in data collection and affects generalizability across different scenarios.
>
> In this study, we explicitly focused on datasets collected through interviews, as this method closely aligns with most clinical scenarios. Both DAIC-WOZ and EATD were collected using an interview process, making them suitable for our analysis and ensuring that our findings are relevant to clinical contexts. Our proposed approach is also designed to handle scenarios where interview questions are not asked in a fixed order of importance, further enhancing its applicability.
>
> While we did not include the visual modality (due to its unavailability in these datasets), our approach achieved competitive results using only audio and the interviewer’s prompts (text modality). This demonstrates the generalizability and robustness of our method across different datasets and interview structures. That said, we recognize that real-world scenarios often introduce additional variability and noise, and improving generalizability remains an important area for future work.
>
>
>
> **Question 2:**
> > Despite the (almost necessary) ablations, the evaluations are fairly superficial, focusing on precision, recall, and F1 but not going deeper into the results to vary the parameters further, or to explain them, or to identify patterns in performance on aspects of the dataset. The result is a fairly ‘paint-by-numbers’ set of experiments with unclear gain in knowledge.
>
> Thank you for your suggestions. In the updated manuscript, we have added a new section, A.7, to discuss the rationale behind using precision, recall, and F1 as evaluation metrics, and to provide a clearer justification for their relevance to this task.
>
> Furthermore, to address the broader aspects of the datasets used in this study, we conducted additional experiments, detailed in Section A.8, to investigate how CoPE and LSR contribute differently to the distinct characteristics of the DAIC-WOZ and EATD datasets. These experiments allowed us to identify patterns in performance and provided insights into which components of our proposed framework are more beneficial and why.
>
> We believe that these additional experiments and discussions significantly enhance the depth of our evaluations, providing a more comprehensive understanding of the contributions and limitations of our approach.
>
> **Question 3:**
> > Could you check your references for, e.g., completeness (e.g., how the WHO 2017 paper was published) or formatting?
>
> Thank you for this detailed observation. We have updated the reference for the WHO 2017 paper to include the correct author name (World Health Organization) and added the URL to the paper in the reference section. This ensures the citation is complete and easily accessible for readers.
>
> **Question 4:**
> > What real-world aspects may be relevant to consider for additional analysis? E.g., how might environmental considerations or more unstructured data collection relate to your results? What are the limitations of this work that you are able to identify?
>
> Thank you for highlighting the importance of considering real-world aspects when implementing our approach in clinical settings. These considerations align closely with the original motivation and purpose of our work.
>
> In the original manuscript, we discussed the limitations of our approach at the end of the paper (Section A.6 in the original submission, now expanded to Section A.10 in the updated manuscript). To address your suggestion, we have further elaborated on the potential impact of environmental factors, unstructured data collection, and variability in real-world scenarios. These expanded discussions aim to identify additional challenges and opportunities for applying our method beyond the controlled settings of curated datasets.
>
> We hope these additions clarify the limitations of our current work and provide a foundation for future research directions.

---

> > ### Comment · Reviewer_eyjC · 2024-11-25
> > **Appendices**
> >
> > Thank you for your response, in particular to questions 1 and 2 with regards to additional information available in the appendices. While I still think some aspects of these should be in the body (with the details naturally in the appendices), my overall recommendation may now be slightly higher, although perhaps only by half a point (or close to one).

---

> > > ### Author Response · Authors · 2024-11-26
> > > **Author Response to Reviewer eyjC (part 2)**
> > >
> > > Dear Reviewer eyjC,
> > >
> > > Thank you for your suggestions. We have added more details to the main body of the text, which you can find highlighted on pages **8** and **10**. Additionally, we have provided guidance for readers to explore further information in the appendix. We made many structural adjustments to accommodate these changes to ensure that the main content still meets the required length.
> > >
> > > We hope these modifications enhance the quality of our work, and we appreciate your consideration in raising the score.

---

> > > > ### Author Response · Authors · 2024-12-01
> > > > **Author Response to Reviewer eyjC (part 3)**
> > > >
> > > > Dear Reviewer eyjC,
> > > >
> > > > Hope you are doing well. Could you please review the changes we have made? If our response addresses the concerns raised, can you consider increasing our overall score? Please let us know if you need any other clarifications. Thank you again for your time!

---

### Official Review · Reviewer_adxY · 2024-11-03

**Soundness:** 3
**Presentation:** 2
**Contribution:** 2
**Rating:** 3
**Confidence:** 4

**Summary:**

The paper presents an architecture for automatic depression detection from clinical interviews. The input information is obtained as a combination of the patient's audio recordings and the interviewer's textual prompts. The key components of the architecture are the Contextual Positional Encoding, which is a combination of absolute and relative positional encodings, and the application of L2 regularization to the fused modality features. The proposed approach achieves competitive results on the DAIC-WOZ benchmark and state-of-the-art results on the EATD benchmark.

**Strengths:**

* The paper does a good job on presenting the current progress on the field of automatic depression detection.
* Ablations are performed for each architectural component.

**Weaknesses:**

* The novelty of the proposed approach comes into question, as it appears a combination of pre-existing methods, applied to well-studied datasets
* There are some issues with the presentation of the method. Specifically, in L257 the equation describing the LSR loss includes a dropout term, while in L340 no such term exists in the loss. My guess is that L257 is inaccurate and simply aims to demonstrate that dropout is applied during training, but this may confuse future readers.
* The presentation of Table 1 could be improved by including some extra columns (Text encoder, Audio encoder etc.)
* No statistical significance testing is presented about the results. This is especially relevant because the explored datasets are fairly small, and the improvements are rather marginal.
* Regarding formatting there are two issues worth of mention. 1) all equations are unnumbered. Including equation numbering can help referencing equations during reviewing or analyzing the paper. 2) The gap after Fig. 1 could be utilized to present more relevant information in the main part of the paper.

**Questions:**

* The Equations about the LSR loss need further clarifications.

---

> ### Author Response · Authors · 2024-11-21
> **Author Response to Reviewer adxY**
>
> Dear Reviewer adxY,
>
> Thank you for your detailed observations and valuable suggestions regarding our presentation. We will respond to the concerns raised one by one.
>
> **Question 1:**
> > The novelty of the proposed approach comes into question, as it appears a combination of pre-existing methods, applied to well-studied datasets
>
> Thank you for raising this concern. While it is true that prior works (as compared in Table 2) have analyzed these datasets using similar modalities—audio from participants and textual prompts from interviewers—none have proposed a solution to address the inherent biases introduced by this approach. Some prior studies have identified the potential for over-reliance on specific question-answer pairs, which improves performance but significantly limits generalizability. However, to the best of our knowledge, no existing work has tackled this challenge directly.
>
> Our study represents the first attempt to address this limitation. By integrating interviewer prompts as an additional textual modality, we improve performance while ensuring robustness and generalizability across different interview structures. Specifically, our framework leverages CoPE and LSR to mitigate biases caused by over-reliance on specific question-answer pairs. This ensures the model remains effective even in scenarios where only random or a limited number of questions are asked.
>
> As a summary, our paper's main novelty and contribution lie in proposing a framework to mitigate the biases introduced by specific prompt-response pairs while maintaining strong performance and generalizability. This focus on improving the robustness of depression detection models represents a meaningful advancement in the field.
>
> **Question 2:**
> > There are some issues with the presentation of the method. Specifically, in L257 the equation describing the LSR loss includes a dropout term, while in L340 no such term exists in the loss. My guess is that L257 is inaccurate and simply aims to demonstrate that dropout is applied during training, but this may confuse future readers.
>
> Thank you for highlighting this inconsistency. You are correct that Dropout is applied during training as a regularization technique and does not explicitly contribute to the LSR loss. To address this, we have removed the Dropout term from Equation 4 and clarified its role as a training-time regularization method. The revised equation now focuses solely on the L2 norm, ensuring consistency with the description provided in Equation 11 later in the manuscript. These changes ensure alignment across the manuscript and avoid potential confusion for future readers.
>
> **Question 3:**
> > The presentation of Table 1 could be improved by including some extra columns (Text encoder, Audio encoder etc.)
>
> Thank you for the suggestion. We have reorganized Table 1 by splitting the results based on the respective datasets to improve clarity. Due to space constraints and page limit, we were unable to add additional columns, but we hope the reorganization enhances the presentation and readability.
>
> **Question 4:**
> > No statistical significance testing is presented about the results. This is especially relevant because the explored datasets are fairly small, and the improvements are rather marginal.
>
> Thank you for highlighting this important aspect. To address this concern, we conducted statistical significance testing on the F1 scores using paired t-tests and Wilcoxon signed-rank tests. The results of these tests, presented in Section A.9 of the Appendix, validate the robustness of our approach and confirm that the observed improvements are statistically significant.
>
> **Question 5:**
> > Regarding formatting there are two issues worth of mention. 1) all equations are unnumbered. Including equation numbering can help referencing equations during reviewing or analyzing the paper. 2) The gap after Fig. 1 could be utilized to present more relevant information in the main part of the paper.
>
> Thank you for your valuable suggestions. We have numbered all equations in the manuscript to facilitate easy referencing and tracking. Additionally, the gap after Figure 1 has been eliminated to better utilize the space and improve the presentation.
>
> **Question 6:**
> > The Equations about the LSR loss need further clarifications.
>
> Thank you for pointing out the need for clarification. We have corrected Equation 4 to address the previous inaccuracies and provided additional clarifications on the LSR loss, including its purpose, formulation, and role in the training process. These updates can be found in Section 3.1, with highlighted changes extending to the end of the section.

---

> > ### Author Response · Authors · 2024-11-27
> > **Author Response to Reviewer adxY (part 2)**
> >
> > Dear Reviewer adxY,
> >
> > Thank you again for taking the time to review our work. With very limited time left to make any further edits to the manuscript, we would like to know if our responses have addressed the concerns raised. If so, could you please consider raising the overall rating? We truly appreciate your feedback, allowing us to improve our work and clarify some ambiguous points with a better presentation.

---

> > > ### Author Response · Authors · 2024-12-01
> > > **Author Response to Reviewer adxY (part 3)**
> > >
> > > Dear Reviewer adxY,
> > >
> > > Hope you are doing well. As the discussion period is approaching its end, we have not yet received response from you. Could you please take a moment to review our response? If we have adequately addressed your concerns, can you consider increasing your overall rating? Additionally, please let us know if you have any further concerns.
> > >
> > > Thank you once again for your time and attention!

---

> ### Comment · Reviewer_adxY · 2024-12-02
> **Reply to authors**
>
> Thanks to the authors for their detailed response and the clear improvements they made on their paper.
> I still have some reservations regarding the technical contribution of this work, therefore unfortunately I cannot update my score.

---

### Official Review · Reviewer_1GMd · 2024-11-04

**Soundness:** 4
**Presentation:** 4
**Contribution:** 2
**Rating:** 5
**Confidence:** 5

**Summary:**

This paper introduces a novel approach to detecting depression from clinical interviews by combining Contextual Position Encoding (CoPE) and Latent Space Regularization (LSR) to overcome limitations of existing methods that rely too heavily on specific question-answer pairs. The system processes both audio responses and interviewer prompts through a multimodal framework, achieving competitive results on the DAIC-WOZ dataset and state-of-the-art performance on EATD.

**Strengths:**

Main contribution/ideas: 1) Proposed to use Contextual Position Encoding (CoPE) to capture the evolving nature of clinical interviews, 2) Uses Latent Space Regularization (LSR) to prevent overfitting and encourage more generalizable patterns, 3) Combines both audio features and interviewer prompts. 1-is the main contribution and should have been the focus of the paper.

The performance is strong compared to other audio and audio+interviewer baselines. Good ablation analysis verifies the importance of CoPE for this dataset

Well-written paper, well-presented ideas, good technical detail.

**Weaknesses:**

I would have personally preferred to focus just on the CoPE embeddings and similar ideas to explore the nature of clinical interviews and how transformers can better model this structures. I think this is the main idea/contribution of the paper and it gets diluted as the authors are chasing SoTA.

Questions about generalizability: I am missing a more detailed deep dive into the databases to understand the impact of each proposed methods to a differently structured interaction, e.g., would CoPE be relevant for a podcast interview, what is the power of representation it is adding to the model etc. Also can you better explain the effect of I, why is it important, is the question order reshufle that you do important?

All methods proposed are not new (cope, lsr, cross-modal attention, feature fusion) so the originality of the paper is limited.

Missing Comparisons with the SoTA multimodal approaches.

**Questions:**

Why don't you present any results even for simple fusion for text and video, since SoTA is what you are after?

See also question above about generalizability and deeper data analysis,

---

> ### Author Response · Authors · 2024-11-21
> **Author Response to Reviewer 1GMd (part 1)**
>
> Dear Reviewer 1GMd,
>
> Thank you for the insightful feedback and concerns regarding novelty and generalizability. Please see our point-by-point response below.
>
> **Question 1:**
> > I would have personally preferred to focus just on the CoPE embeddings and similar ideas to explore the nature of clinical interviews and how transformers can better model this structures. I think this is the main idea/contribution of the paper and it gets diluted as the authors are chasing SoTA.
>
> Thank you for this thoughtful suggestion. As discussed in the fourth paragraph of the 'Introduction' section, one key characteristic of clinical interviews is that questions are typically asked in a structured order, starting with general inquiries (e.g., "Where do you come from?" or "How did you get here today?"), progressing to personal topics such as occupation and family status, and culminating in mental-state probing questions. Previous studies have leveraged this structure to achieve highly competitive results, often by focusing on later-stage, mental-state probing questions while neglecting earlier, seemingly less relevant parts of the interview.
>
> While this approach has yielded strong performance, it is not without limitations. It relies on exploiting specific question-answer pairs, a pattern that is not always available in real-world scenarios. For instance, in the EATD dataset, the questions asked are completely random. Moreover, disregarding earlier portions of the interview risks losing critical contextual cues, which can be especially detrimental given the scarcity of labeled data in this domain.
>
> To address these challenges, we introduced CoPE and LSR to model the sequential and contextual dynamics of the entire interview, rather than selectively focusing on later-stage data points. Our approach ensures that meaningful information is captured across the full conversation, making it more robust and generalizable across datasets. For example, in the DAIC-WOZ dataset, we intentionally did not follow prior approaches that selectively focus on hand-picked question-answer pairs to achieve SoTA. Instead, we prioritized designing a method that is broadly applicable and not overly reliant on dataset-specific patterns.
> We hope this clarification further highlights our original motivation and the unique contribution of this work. Let us know if additional details are needed.
>
> **Question 2:**
> > Questions about generalizability: I am missing a more detailed deep dive into the databases to understand the impact of each proposed methods to a differently structured interaction, e.g., would CoPE be relevant for a podcast interview, what is the power of representation it is adding to the model etc. Also can you better explain the effect of I, why is it important, is the question order reshufle that you do important?
>
> Thank you for these valuable questions about generalizability and the importance of different components of our model.
> In this work, we used two datasets with distinct interview structures to evaluate the robustness of our approach:
>
> DAIC-WOZ: This dataset contains long interviews (10+ minutes on average) conducted between a virtual interviewer and participants, providing rich sequential and contextual information.
>
> EATD: This dataset consists of much shorter interactions, where participants are asked only 3 questions, often lasting less than a minute.
>
> To analyze the performance of our model across different interaction lengths, we conducted additional experiments, detailed in Section A.8. These results demonstrate that CoPE effectively captures contextual dependencies across varying interaction lengths, making it applicable to both structured, lengthy interviews and shorter, less structured interactions. While CoPE was specifically designed for the structure of clinical interviews, its ability to encode positional and sequential information suggests potential applicability to other settings, such as podcast interviews, where conversational flow plays a critical role.
>
> The inclusion of the interviewer’s prompt as a text modality provides two key advantages:
>
> **Semantic Information:** The prompts convey the meaning of the questions, which helps the model understand the context of the participant’s response.
>
> **Order Information:** The prompts implicitly encode the sequential structure of the interview. This order information, combined with acoustic cues, helps the model better identify "triggering" questions that may elicit responses indicative of a participant’s mental state.
>
> Compared to prior approaches that focus solely on mental-state-specific questions or hand-picked question-answer pairs, our model considers the entire interaction. This ensures robustness, even if question-answer pairs are reshuffled or the questions themselves are not directly mental-state related, as demonstrated by our experiments on the EATD dataset, where the questions are completely random.

---

> > ### Author Response · Authors · 2024-11-21
> > **Author Response to Reviewer 1GMd (part 2)**
> >
> > **Continue for question #2**
> >
> > To assess the quality of the learned feature representations, we utilized t-SNE to visualize the high-dimensional fused features produced after modality-specific processing and attention-based fusion. The visualization, included in Section A.9, revealed distinct clusters separating depressed and healthy participants on both datasets. This demonstrates that the combination of CoPE and attention mechanisms effectively integrates multimodal information, creating meaningful and separable representations before classification. These results highlight the power of CoPE in generating robust, contextually aware representations that generalize across different datasets and interaction structures.
> >
> > **Question 3:**
> > > All methods proposed are not new (cope, lsr, cross-modal attention, feature fusion) so the originality of the paper is limited.
> >
> > Thank you for raising this concern about the novelty of our approach. The downstream task we address in this work—multimodal depression detection as a binary classification problem—has indeed been widely studied, especially over the past decade. However, as highlighted in the 'Introduction' section of our manuscript, it is only recently that researchers have started to recognize the benefits of integrating the interviewer’s prompts as a textual modality to improve prediction performance.
> >
> > Previous studies have shown that models leveraging specific question-answer pairs achieve significant improvements, but these findings also reveal a critical limitation: such approaches lack generalizability. When interview questions are shuffled or randomized (e.g., as in the EATD dataset), or when only generic questions are asked, these models suffer significant performance drops. To the best of our knowledge, this issue has not been explicitly addressed in prior work.
> >
> > It is true that we did not develop entirely new algorithms, our contribution lies in addressing this gap. By integrating the interviewer’s prompts as a textual modality and designing a framework that avoids overfitting to specific question-answer pairs, we ensure robustness and generalizability across datasets with varying interaction structures. Our proposed use of CoPE and LSR builds upon established methods, but we adapt and apply them in a novel way to address the unique challenges of depression detection in clinical interviews. This combination of innovation in problem framing and practical implementation represents the core originality of our work.
> >
> > We hope this explanation clarifies the significance of our contributions. Let us know if further details are required.
> >
> > **Question 4:**
> > > Missing Comparisons with the SoTA multimodal approaches.
> >
> > Thank you for raising this concern. In Table 2, we listed the SoTA approaches for the DAIC-WOZ and EATD datasets. Although our study does not incorporate visual information, we included prior approaches that used the vision modality to provide a comprehensive comparison. To ensure fairness and consistency, we adopted the same evaluation metrics as these studies, which are also the most commonly used metrics for this task.
> >
> > If there are additional multimodal approaches on either dataset that we may have overlooked, we would greatly appreciate your suggestions so we can include them in future revisions for a more thorough comparison.
> >
> > **Question 5:**
> > > Why don't you present any results even for simple fusion for text and video, since SoTA is what you are after?
> >
> > Thank you for pointing this out. The primary reason for not including video data in this work is that it is not available in either of the datasets used. For the DAIC-WOZ dataset (requested from https://dcapswoz.ict.usc.edu/), the original data collection included video recordings. However, the version of the corpus we received contains only audio recordings and text transcriptions. For the EATD dataset, no video data was collected during its data collection phase, so this modality is unavailable.
> >
> > We have discussed this limitation and the rationale behind our approach in more detail in Section A.10 of the Appendix. Please let us know if additional clarification is needed.
> >
> > **Question 6:**
> > > See also question above about generalizability and deeper data analysis.
> >
> > Thank you for your suggestions regarding deeper data analysis. In response, we have conducted additional experiments, which are detailed in Sections A.6 and A.8 of the Appendix. We hope this deeper analysis provides greater insights and strengthens the overall contribution of our work.

---

> > > ### Comment · Reviewer_1GMd · 2024-11-26
> > >
> > > I wish to thank the authors for the detailed reply and the (clearly demarcated) updates in the paper. The paper has improved, however, the main weakness (limited contribution) remains - so I will not update my overall rating.

---

> > > > ### Author Response · Authors · 2024-11-26
> > > > **Author Response to Reviewer 1GMd (part 3)**
> > > >
> > > > Dear Reviewer 1GMd,
> > > >
> > > > Thank you for acknowledging our efforts to address your concerns and improve the quality of our work. In our responses to questions **1**, **2**, and **3**, we provided more detailed explanations and conducted additional experiments. We outlined the challenges researchers encounter in addressing this task, our motivation, and our distinct method for approaching it, along with its generalizability, originality, and our specific contributions.
> > > >
> > > > We appreciate your insights regarding our contributions, and we understand your concerns. We are grateful for the time you have taken to review our work during this busy period. **If there is anything else we can do to improve the overall rating, we would love to know.** Thank you for your consideration!

---

### Official Review · Reviewer_SKAj · 2024-11-09

**Soundness:** 2
**Presentation:** 3
**Contribution:** 2
**Rating:** 6
**Confidence:** 4

**Summary:**

The document presents a novel approach to detecting depression by leveraging multimodal data from clinical interviews. This method combines **Contextual Position Encoding (CoPE)** and **Latent Space Regularization (LSR)** to enhance model accuracy and generalizability. By analyzing both participants' audio responses and interviewers' text prompts, CoPE enables the model to capture the full conversational context rather than relying on isolated cues. Meanwhile, LSR helps prevent overfitting by encouraging the model to focus on meaningful patterns in the data. The applicability of the proposed approach has been evaluated using two state-of-the-art datasets, including **DAIC-WOZ** and **EATD**.

**Strengths:**

* Multimodal approach relying on the inclusion of contextual positional embeddings and latent space regularization for depression detection.
* Inclusion of participant's audio responses and the interviewer's text prompts for improving the accuracy and generalizability of depression detection models.
* State-of-the-art results on the DAIC-WOZ and EATD datasets for multimodal depression detection.

**Weaknesses:**

* The interview is classified as a distinct modality; however, it would be more accurately rephrased as a text modality.
* In **Table 1**, it is unclear whether CoPE has been applied to the text modality within the first set of audio-specific models.
* Why was the visual modality not considered in the modeling phase?
* In **Table 2**, the table should be divided into sections based on respective datasets.
* LSR implements regularization at the level of latent embeddings. How does this compare to weight-based approaches, such as L2 and L1 regularization techniques?
* Instead of utilizing attention-based weighting, what was the rationale behind not considering cross-attention-based fusion for the text and audio modalities?

**Questions:**

* Have the authors thought about utilizing RoPE embeddings for encoding relative positions regarding contextual position encoding?
* Do the existing contextual CoPE embeddings apply effectively to interview sessions of varying lengths, from short to long?

---

> ### Author Response · Authors · 2024-11-21
> **Author Response to Reviewer SKAj (part 1)**
>
> Dear Reviewer SKAJ,
>
> Thank you for your feedback and all the valuable and constructive questions and suggestions. Let us respond to your concerns point by point.
>
> **Question 1:**
> > The interview is classified as a distinct modality; however, it would be more accurately rephrased as a text modality.
>
> Thank you for pointing this out and we totally agree with it. In the original manuscript (e.g., lines 023, 096, 203, and 394), we clarified that the interview data is represented as a text modality. In Table 2, we used the label ‘I’ to denote ‘Interview’ to distinguish it from prior approaches that primarily utilized transcribed text from participants' responses as the text modality.
> To enhance clarity, we have revised lines 014–016 in the Abstract to explicitly state this distinction.
>
> **Question 2:**
> >In Table 1, it is unclear whether CoPE has been applied to the text modality within the first set of audio-specific models.
>
> Thank you for pointing out this ambiguity. In Table 1, we created baseline models for both unimodal (audio-only) and multimodal approaches. CoPE has only been applied where explicitly indicated under the "Model" column. As such, CoPE was not applied to the text modality in any of the audio-specific models, except in the 'Full model.' To address this ambiguity, we have revised the caption of Table 1 to clarify this distinction.
>
> **Question 3:**
> > Why was the visual modality not considered in the modeling phase?
>
> Thank you for raising the consideration of the visual modality. The primary reason it was not included in the modeling phase is that the datasets used in our study contain only audio data and text, with no video data available. Specifically:
>
> For the DAIC-WOZ dataset, while video recordings were collected during the original data collection, the version provided to us upon formal request included only audio recordings and transcriptions of the interactions between the interviewer and participants.
>
> For the EATD dataset, no visual data was collected during its data collection phase; it is limited to audio and text data.
>
> As briefly mentioned in the limitations section of the Appendix (Section A.10), we recognize that incorporating features from additional modalities, such as vision or physiological data, could potentially enhance prediction performance. We acknowledge the potential benefits of such multimodal integration and intend to explore these possibilities as more comprehensive datasets become available.
>
> **Question 4:**
> >In Table 2, the table should be divided into sections based on respective datasets.
>
> Thank you for the suggestion. We have redesigned and reorganized Table 2 to group results by their respective datasets, improving clarity and readability. Additionally, to maintain consistency in presentation, we have updated Table 1 to follow the same structured format. Please refer to the updated manuscript for the revised tables.
>
> **Question 5:**
> >LSR implements regularization at the level of latent embeddings. How does this compare to weight-based approaches, such as L2 and L1 regularization techniques?
>
> Thank you for highlighting the comparison with weight-based regularization techniques. To address this, we have expanded the discussion in Section 3.1 to elaborate on the differences between Latent Space Regularization (LSR) and traditional weight-based approaches such as L2 and L1. Additionally, we conducted further experiments to evaluate the performance of LSR against these techniques, with results presented in Section A.6 of the Appendix. These updates provide a comprehensive analysis of LSR and its advantages over weight-based regularization in the context of our study.
>
> **Question 6:**
> >Instead of utilizing attention-based weighting, what was the rationale behind not considering cross-attention-based fusion for the text and audio modalities?
>
> Thank you for this insightful comment. The objectives and unique characteristics of our task primarily guided our decision to use attention-based weighting rather than cross-attention-based fusion.
>
> Our framework is designed to maintain modularity and interpretability in fusing text and audio features. By employing attention-based weighting, we independently extract the most relevant features from each modality before combining them, ensuring a straightforward interpretation of how each modality contributes to the final fused representation. This approach also preserves modality-specific information, which is critical for understanding the distinct roles of text and audio in our task.

---

> > ### Author Response · Authors · 2024-11-21
> > **Author Response to Reviewer SKAj (part 2)**
> >
> > **Continue for Question #6**
> >
> > In contrast, while potentially offering improved performance, cross-attention-based fusion introduces additional complexity and computational overhead due to the pairwise interactions between features from both modalities. Our chosen attention-based weighting mechanism strikes a balance between computational efficiency and robust performance for our application, which involves large-scale multimodal data. By emphasizing the most informative aspects of each modality, we avoid the risk of noisy or diluted representations that can sometimes result from cross-modal interactions.
> >
> > Additionally, integrating CoPE within our independent attention mechanism enhances contextual relevance without necessitating complex cross-modal dependencies. This design choice allows for the flexible adaptation and fine-tuning of attention weights for each modality, aligning with our framework’s focus on efficient, contextualized multimodal fusion.
> >
> > While we acknowledge the merits of cross-attention-based fusion, we believe that our current approach's modularity, interpretability, and computational efficiency are well-suited to the goals and constraints of our framework. As part of future work, we are open to exploring cross-attention mechanisms to assess their impact on performance and interpretability.

---

> > > ### Comment · Reviewer_SKAj · 2024-11-25
> > > **Reviewer response**
> > >
> > > Thanks to the authors for the detailed response. While all the major points have been addressed in the response, it would be great if the authors could mention why relative positional encoding schemes were not used (like RoPE).
> > > Based on the response, I would like to increase the rating by 1.

---

> > > > ### Author Response · Authors · 2024-11-26
> > > > **Author Response to Reviewer SKAj (part 3)**
> > > >
> > > > Dear Reviewer SKAj,
> > > >
> > > > Thank you once again for bringing up relative positional encoding (**RoPE**). After reviewing our original response, we realized that it was not included in our first response despite being saved in our local document. We apologize for the oversight and any inconvenience caused. We have made some revisions to the text, and please see the updated version below.
> > > >
> > > > RoPE focuses on encoding the relative positions of sequence elements instead of their absolute positions. This approach is particularly effective for tasks where the relationships between elements are more important than their specific locations. RoPE represents positional information as the relative distances between tokens, utilizing a rotational transformation. For example, rather than stating "this token is at position 3," RoPE emphasizes that "this token is 2 steps away from token 1." This representation is achieved through sinusoidal embeddings combined with rotation matrices.
> > > >
> > > > **Example:** For the sentence "I feel very sad": Absolute positions are "I" (1), "feel" (2), "very" (3), "sad" (4). Relative positions mean "feel" is 1 step away from "I," and "sad" is 2 steps away from "feel."
> > > >
> > > > RoPE encodes relationships, such as one step or two steps, and adjusts the embeddings accordingly. This makes it effective for tasks that involve flexible ordering or where pairwise relationships are particularly important. However, a **limitation** of RoPE is that it may not effectively capture task-specific hierarchical contexts.
> > > >
> > > > In contrast, Contextual Position Encoding (CoPE) was chosen for our work because it combines both absolute positional information and contextual weights tailored for task-specific encoding. CoPE is especially suitable for tasks with hierarchical or task-specific dependencies, such as clinical interviews. For example:
> > > > It encodes the specific position of a token (e.g., words in a sentence or the sequential order of questions in an interview).
> > > > It also encodes the relative importance of the token within the broader context of the task.
> > > >
> > > > This dual focus aligns well with the structure of clinical interviews, where the sequence and significance of questions differ, making CoPE more effective in capturing the flow and context of interactions.
> > > >
> > > > We acknowledge that RoPE could complement CoPE in our task. For example, RoPE may improve pairwise dependency modeling, such as comparing specific question-answer pairs, but it might struggle to capture hierarchical importance. Exploring RoPE as a supplementary encoding mechanism alongside CoPE could enhance the encoding of nearby interactions and dependencies, especially in tasks where local relationships are crucial.
> > > >
> > > > We greatly appreciate your suggestion, as it provides valuable insights and directions for our future work. We hope this explanation helps address your concern.

---

> > > > > ### Comment · Reviewer_SKAj · 2024-11-26
> > > > > **Reviewer Response**
> > > > >
> > > > > Thanks to the authors for the response regarding RoPE and the fine-grained differences with CoPE.
> > > > > The idea of using CoPE along with RoPE for pairwise dependency modeling is really interesting.

---

> > > > > > ### Author Response · Authors · 2024-11-26
> > > > > > **Author response to reviewer SKAj (part 4)**
> > > > > >
> > > > > > Dear Reviewer SKAj,
> > > > > >
> > > > > > Thank you for taking the time to review our work. We appreciate your insights on RoPE, which gives us additional perspective on this task and related topics. We are also very grateful for your decision to raise the overall rating.

---

### Author Response · Authors · 2024-11-21
**Author response & revision**

Dear reviewers, ACs, and PCs,

We appreciate the time you have taken to review our work and for giving us the opportunity to address the concerns raised and improve the quality of our paper 'Multimodal Depression Detection with Contextual Position Encoding and Latent Space Regularization.' We have summarized our responses to each reviewer’s concerns for easier tracking and review, including the changes made in the **updated pdf file** and **additional experiments** conducted.

We have just uploaded the revised version with **highlighted changes** for your review and will upload a **clean version** before the discussion period ends. During the discussion period, we welcome any further feedback from the reviewers. Thank you again!

Following Reviewer SKAj’s suggestions:

1.	Clarifications in Abstract.

2.	Updated the caption of Table 1.

3.	Experiments in Section A.10 and discussed via response.

4.	Restructured Table 2 and Table 1.

5.	Updates in Section 3.1 and experiments in Section A.6.

6.	Discussed via response.

7.	Discussed via response.

8.	Added Figures 3 and 4, experiments in Section A.8, and discussed via response.



Following Reviewer 1GMd’s suggestions:

1.	Discussed via response

2.	Discussed via response

3.	Experiments in Section A.8, Section A.9

4.	Discussed via response

5.	Discussed via response; Section A.10

6.	Experiments in Section A.6, Section A.8


Following Reviewer adxY’s suggestions:

1.	Discussed via response

2.	Section 3.1

3.	Table 1

4.	Experiments in Section A.9

5.	All equations (numbered); extra space eliminated

6.	Section 3.1


Following Reviewer eyjC’s suggestions:

1.	Discussed via response; Section A.10

2.	Discussed via response; experiments in Section A.7, Section A.8

3.	Reference fixed

4.	Discussed via response; Section A.10

---

### Meta-Review · Area_Chair_ypFS · 2024-12-20

**Metareview:**

This paper presents a multimodal model for automatic depression detection from clinical interviews which combines the patient's audio recordings with the interviewer's textual prompts. The authors propose a Contextual Positional Encoding combining both absolute and relative positional encodings, and the application of a smoothing regularization term to the latent space to enforce consistency in the model’s learned representations. The proposed approach achieves competitive results on the DAIC-WOZ benchmark and state-of-the-art results on the EATD benchmark.

3 reviewers were negative about the paper, and one gave marginal above acceptance. All reviewers found the problem setting interesting, that the performance was strong, and the ablation studies and analysis were solid.

However, all reviewers were also concerned that the proposed methods and combination of modules (contextual positional encodings, latent space regularization, cross-modal attention, feature fusion) already exist in the literature, so they found limited originality in the paper. I am aligned with the majority of the reviewers' judgement that the application of these methods to this new problem of multimodal depression detection warrants research, but that the application-centric and practical tricks proposed here are too specific to be broadly significant for the ML community, and therefore the paper falls below the threshold for acceptance.

**Additional Comments On Reviewer Discussion:**

Based on the rebuttal, reviewer SKAj raised their score from 5 to 6, including a long discussion regarding the benefits of relative PE and contextual PE.

However, all other reviewers were not convinced, and remained negative. In particular, the authors argued that multimodal depression detection integrating the interviewer’s prompts as a textual modality with the patient's audio recordings is a new problem, and their work should be perceived as applying existing techniques to fill the application gap with some practical tricks to improve generalization. The reviewers did seem to appreciate this point but unfortunately none felt strongly that these contributions were significant enough for the broader ML community.

---

### Decision · Program_Chairs · 2025-01-22

Reject